# ISO 22000 Certification: Diffusion in Europe

Nathalia Granja [1,2,*], Pedro Domingues [3], Mónica Cabecinhas [3], Dominik Zimon [4] and Paulo Sampaio [3]

1   School of Engineering, University of Minho, 4710-057 Braga, Portugal
2   Faculty of Sciences, University of Porto, 4099-002 Porto, Portugal
3   ALGORITMI Research Centre, Department of Production and Systems Engineering, University of Minho, 4710-057 Braga, Portugal; pdomingues@dps.uminho.pt (P.D.); monica.cabecinhas@gmail.com (M.C.); paulosampaio@dps.uminho.pt (P.S.)
4   Department of Management Systems and Logistics, Rzeszow University of Technology, 35-959 Rzeszów, Poland; zdomin@prz.edu.pl
*   Correspondence: nathaliagranja@gmail.com

**Abstract:** The main aim of this paper is to answer the research question, "Is the Gompertz model suitable for studying the diffusion of the ISO 22000 standard in Europe?" Forecasting models adopting the Gompertz model were developed to estimate to which extent the Food Safety Management Systems (FSMS) based on the ISO 22000 standard are expected to be implemented and certified in the European continent. To provide a forecast for the next few years, data from the diffusion of renowned ISO standards, namely, ISO 9001 and ISO 14001, were extrapolated in order to overcome the shortcoming since data concerning ISO 22000 certification was only available for the period 2007 to 2018. The results provide a cross-sectional portrayal of the European diffusion of ISO 22000 certification and suggest an interval of the estimated number of certificates issued in Europe. This research paper presents the first attempt to empirically analyze the dynamic of diffusion of the European ISO 22000 certification. A more accurate fitting with real results may be expected with further information available in the forthcoming years.

**Keywords:** certification; food industry; international standards; modeling





## 1. Introduction

ISO 22000 certification can be adopted by any organization directly or indirectly involved in the food chain, encompassing both consumer and market needs [1]. Related to this, company concerns associated with new requirements from stakeholders have increased over the last few years. This competitive context outlines the path to adopting international standards and consequently to earning the trust of partners and final consumers [2].

It is important to evaluate the ISO standards data for Europe, as it constitutes an important geostrategic region for the global economy and quality. Europe represents a positive contribution, accounting for 31% of global ISO 22000 certification [3].

Moreover, the diffusion of food safety management standards is a current need due to the fact that food losses have an impact on food quality and safety [4]. This is a fundamental issue because the modern world is struggling with a huge problem of food waste [5]. Nowadays, food safety is an evident concern among the consumers of sustainable agricultural products, which should address both farmers' and consumers' welfare [6]. Recent global events evidence that our current food production is unsustainable, and issues related to the increasing population, increased food demand, and high levels of food waste in the food systems emphasize the need for more sustainable practices [7]. This problem is so significant that the UN Sustainable Development Goals include "Ensure sustainable consumption and production patterns" with "Halve per capita global food waste at the retail and consumer levels and reduce food losses along production and supply chains, including post-harvest losses" [8].



Specifically, in the food industry, another constantly growing concern is the quality of food products regarding safety. Furthermore, besides responsibility for producing safe food, these industries must evidence that food safety is planned and assured [9]. The risk of incidents related to food safety hazards is real, since food products are produced using different techniques and transported all over the world [10].

ISO 22000 is an international standard that requires food safety management system implementation to help address food safety requirements and growing trends like food security and sustainability in the food chain. The latest revision of the ISO 22000 standard (in 2018) is in line with the ISO 9001 standard, which aims to provide a better understanding of the concept of operational and strategic level of risk and includes improvements to definitions, in line with Codex Alimentarius [11].

Moreover, many authors emphasize that the ISO 22000 standard can support the proper implementation of important processes in both enterprises ([12,13]) and food supply chains ([14,15]). Therefore, ISO 22000 FSMS not only allows for the growth of food safety but also enables the improvement of key processes in the food supply chain, especially in production and control processes [16]. It should be emphasized that the dissemination of standards supporting the care of food quality at every stage of the supply chain is currently a necessity.

When reviewing the literature, a research gap emerged between the literature that analyzes the diffusion of ISO 22000 standard and the diffusion of the renowned ISO 9001 standard or other international standards, such as ISO 14001. With the above in mind, the main focus of the article is to analyze ISO 22000 certification behavior in Europe based on available data as well as the widely studied diffusion process of the aforementioned ISO management standards since implementation of ISO food safety management aids organizations to manage safety hazards while working together with other ISO management standards, such as ISO 9001 [11]. Developed forecasting models helps to estimate to which extent the FSMS based on the ISO 22000 standard are expected to be implemented and certified in the European continent.

The main goal of this work is to answer the following research questions: Is the Gompertz model suitable for studying the diffusion of ISO 22000 in Europe? Can we consider the diffusion of widely adopted ISO standards, such as ISO 9001 and ISO 14001, to study the diffusion of ISO 22000? Has Europe already reached the saturation stage of certification?

Therefore, this paper reflects on what we believe to be a pioneering contribution to reporting the diffusion and a forecasting model throughout the years of European FSMS encompassing the ISO 22000 standard.

This paper is structured as follows: The Introduction section is followed by the Literature Review section, which address and dissects the latest published contributions concerning implementation and diffusion of FSMS standards as well as the main attributes that impact this dissemination. The research methodology is described in the following section and the Findings section presents an analysis of the evolution of ISO 22000 throughout the years, considering the number of certificates issued in Europe. Finally, the conclusions, suggestions for future work, and references used in this research are presented.

## 2. Literature Review

Issues concerning the increasing population, increased food demand, and high levels of food waste in the food systems entail the need for more sustainable practices [7]. Consumers are very peculiar about food safety and the newly health-conscious are raising concerns about sustainable products and practices in the food chain [6].

Food loss and food waste in the food chain is a global issue, irrespective of region or the economic income level [17]. Related to this, global food systems have a significant environmentally friendly contribution at all stages of the food chain [18]. The level of loss and waste is reduced in companies where the processes are running in harmony with certain standards [19].

The diffusion of FSMS is a current need since food losses have an impact on food quality and safety [4]. According to the Food and Agriculture Organization of the United Nations (FAO), in the food chains of high-income countries, waste is mostly generated in the consumer phase. On the other hand, more than 40% of the food losses in low–middle-income countries occur at the production-to-processing stages of the food supply chain [4] because many of these countries lack the infrastructure to properly store food [17]. These data may be related to the dissemination of FSMS certification, whereas per-capita GDP has a positive impact on a country's certification process. To demonstrate that, most private food safety standards are based in the United States and Europe [20].

Moreover, in a study carried out by Zimon and Domingues in 2020 [16] concerning the impact of the implementation of ISO 22000 on food safety, the vast majority of surveyed organizations recognized the implementation of the aforementioned standard as an added value. Benefits of implementation in that study were classified, such as demonstrating compliance with legal requirements and food safety regulations, producing and delivering products that are safe for customers, and increasing staff awareness of its impact on food quality and safety.

Based on the Codex principles for food hygiene (including the HACCP System—Hazard Analysis and Critical Control Point), ISO 22000 is an international standard that requires an FSMS implementation to assure food safety and to support sustainability in the food chain. Moreover, this FSMS is in line with ISO 9001 in the latest revision of ISO 22000:2018, and that improvement provides a better understanding of the concept of operational and strategic level of risk [11] and also continues to support a company to be transparent about the control of food safety risks due to compliance with the requirements of this standard [13].

In addition to ISO 22000, the international benchmarking institution GFSI (Global Food Safety Initiative) recognizes some food safety standards such as Food Safety System Certification 22000 (FSSC 22000), the British Retail Consortium Food Standard (BRC), the Global Good Agricultural Practices (GlobalG.A.P.), and International Featured Standards Food (IFS Food). The aforementioned private food safety certifications encompass an important and influential regulatory mechanism related to the contemporary agri-food system [20].

Concerning the spread of standards, it should be mentioned that they present very different geographical coverage. Analyzing the cross-national adoption of six major private food safety standards, Mohammed and Zheng in 2017 [20] considered that these standards are adopted mainly by their own and surrounding countries, whereas some others have a much wider international adoption. The authors mentioned that European countries have the highest number of certificates obtained from the four European standards (Table 1). However, FSSC 22000 and ISO 22000 are widely adopted by countries outside Europe, such as China and India.

**Table 1.** European countries from the top 10 countries by the number of certified sites considering different food safety standards in 2013 (adapted from Mohammed and Zheng [20]).

| Country | BRC | FSSC 22000 | GlobalG.A.P. | ISO 22000 |
|---|---|---|---|---|
| Belgium | 585 | - | 3185 | - |
| Germany | 603 | 299 | 9008 | - |
| Greece | - | - | 11,367 | 1720 |
| Italy | 2328 | - | 20,218 | 781 |
| Netherlands | 1312 | 367 | 8625 | - |
| Poland | 796 | - | 3163 | 640 |
| Romania | - | - | - | 1014 |
| Russia | - | 263 | - | - |
| Spain | 1551 | - | 32,149 | 525 |
| Turkey | 537 | - | - | 733 |
| United Kingdom | 3786 | - | - | - |

The aforementioned authors evidenced that the number of domestic certification bodies is also an important aspect concerning the adoption of food safety certification. It may help to understand the pattern of GlobalG.A.P. adoption, since this trademark (focused on good agricultural practices) currently has 163 approved certification bodies around the world [21].

Since health-conscious consumers are involved in issues from food safety to environmental protection [6], implementing a proper quality management system, together with legal measures and including management of the waste generated, contributes to raising awareness and is the key for the future sustainable development [22]. Therefore, the consequences of producing food that does not meet the food safety requirements can be very serious [23]. In addition to food loss and waste, not complying with minimum requirements of food safety management standards can lead to food being unsafe, such as the production of toxins in food itself, the use of contaminated water, and inadequate and unhygienic storage conditions [4]. That is why researchers such as Gaaloul et al. [24] and Zimon and Domingues [16] emphasized that food safety management standards are a proper tool to aid organizations in identifying and controlling food safety hazards.

ISO 22000 certification can be adopted by any organization directly or indirectly involved in the food chain, encompassing both consumer and market needs [1]. This standard maps out what the organizations should implement to demonstrate their ability to control food safety hazards in order to ensure that food is safe [11]. Organizations that hold a certified management system compliant with ISO 22000 can guarantee their recipients products that are manufactured, transported, and stored following the highest safety rules [25]. Furthermore, organizations recognize that implementation and improvement of quality management systems are positive for operating effective and efficient supply chains [26]. In a study addressing quality management systems and the effectiveness of food supply chains, the abovementioned author stated that increased efficiency and effectiveness occur when quality management systems are supported by carefully selected strategies.

Concerning the security of the global food supply chain, the ISO 22000 standard allows for a more efficient risk management and is an effective tool to understand the requirements for organizations operating throughout the food chain [27]. It should be emphasized that the situation in the market and the expectations of businesses and consumers have changed significantly over the last 10 years. Meanwhile, the companies encompassing the food chain have been faced with the need to meet new food safety requirements.

Another point to be stressed is the globalized food supply chain. Nowadays, food items are produced, transformed, and consumed in very different parts of the world and the impact of growing international trade on food losses still needs to be properly assessed [4]. Furthermore, due to the globalization of trade and international trade in food products, it is also necessary to comply with international management standards [10]. One should bear in mind that the current global food supply encompasses different types of intermediaries who operate under a diverse range of food quality and safety regulations [28].

In accordance with this, the number of ISO 22000 certifications has increased during the last decade in the food industry (Figure 1), reflecting not only the required food quality, but also the aspiration of these companies to improve their image in the domestic market and to access foreign markets, since ISO 22000 certification is a potential marketing tool due to it being a common language with stakeholders [29]. As with the ISO 9001 standard, the common language properties can minimize the communication frictions endemic to trade between firms from different countries by allowing for effective communication of internal production systems [30].

It should be noted that the main objective of ISO 22000 certification is to globally harmonize food safety management arrangements among food chain organizations [16] since ISO's food safety management standards work together and also with other renowned ISO management standards, such as ISO 9001 [11].

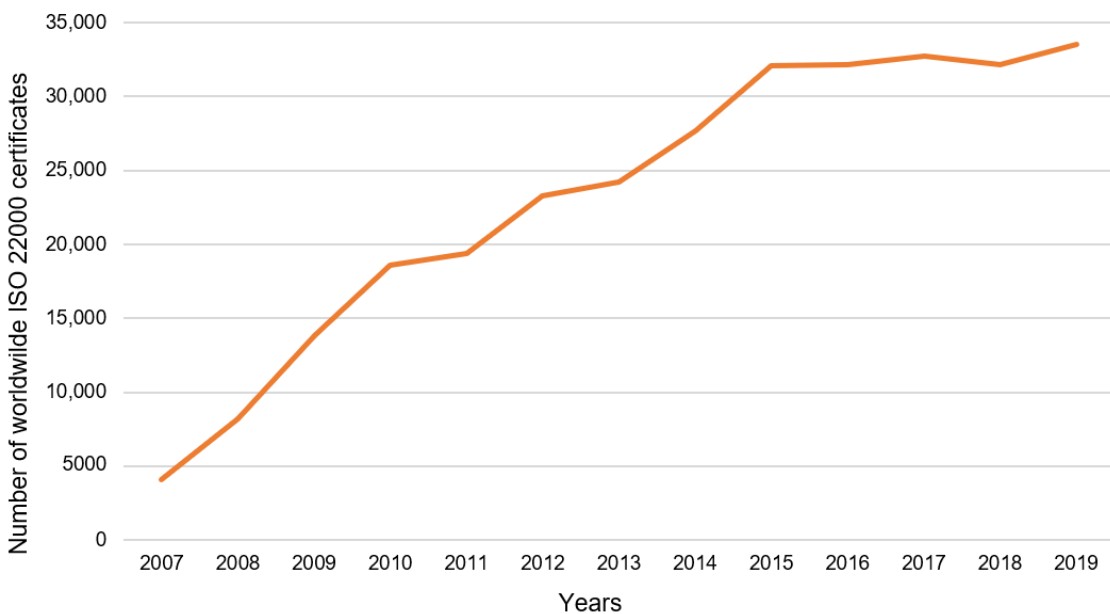

**Figure 1.** Evolution of the worldwide ISO 22000 certificates.

ISO 22000 certification also provides assurance within the global supply chain, allowing products to cross borders and increasing the confidence of customers [11]. To demonstrate that, El-Rouby et al. [31] reported the implementation of the ISO 22000 food safety management system in a spaghetti industry that outputted enhanced food protection, increased customer trust, and improved cost efficiency.

Analyzing the impact of ISO 22000 certification in a food business, Toufaili and Halawi [32] concluded that the implemented FSMS also positively impacts many aspects, such as ISO 22000 changing job requirements, enhancing job performance, improving productivity and quality of work, affecting employee commitment, and increasing employee motivation. In agreement with this, in a multiple case study of the performance of FSMS, in which specific outputs of non-ISO 22000 certified outlets were better than outlets that were ISO 22000 certified, Fathurrahman et al. [33] stated, recently, that successful implementation of FSMS depends both on system performance and on the context wherein it operates.

Other studies identified and summarized the main obstacles, two of which are financial constraints [34] and certification costs [35]. All these difficulties impact the decision to implement and certify an ISO 22000 FSMS due to the necessity to invest money, time, and organizational resources [29].

Despite the obstacles mentioned above, ISO 22000 FSMS not only allows for the growth of food safety but also allows for the improvement of key processes in the food supply chain, especially in production and control processes ([15,16]). Updating the implementation of ISO 22000:2018 and HACCP in a mushroom product manufacturer, Chen et al. [36] reported a statistically significant decrease in the anomalies of process flow in the first year, meeting the requirements of this standard and minimizing the occurrence of problems such as product contamination and food hazards. Therefore, implementation that addresses the requirements of the ISO 22000 standard not only contributes to reducing the number of errors, but also to reducing food waste and loss in food supply chains [15].

Nevertheless, food trade is headed by private standards required by large retailers, some wholesalers, and foodservice companies [37], and the certification of these standards recognized by the GFSI allows for the improvement of reputation and competitive advantage [29].

ISO 22000 is not by itself a GFSI-recognized FSMS, but through the combination with other standards and technical specifications for sector PRPs (prerequisite programs), it

encompasses the Food Safety System Certification 22000 [29]. The FSSC 22000 scheme was recognized by GFSI in 2010 and is based on big manufacturers that aim for "a common set of prerequisite programs that can be used by any manufacturer who wishes to establish an ISO 22000 certified FSMS" [38]. Since ISO 22000-certified companies are transitioning and requiring the upgrade to FSSC certification, this international recognition represents a real issue in the food industry.

Considering the literature review carried out by Teixeira and Sampaio [39], there are some studies related to FSMS and ISO 22000. However, in opposition to the renowned ISO 9001 standard and other international standards, such as ISO 14001, studies related to the diffusion of ISO 22000 are scarce. However, the dissemination of standards and certification schemes globally is relatively limited and may result in reduced market access [40]. Related to that, Clougherty and Grajek [30] stated that diffusion of the ISO 9001 standard evidences a trade barrier for lower–middle-income countries, whereas high-income countries hold most of the benefits from worldwide standardization. The stricter requirements of food safety standards imposed by developed countries could limit food-export processes from developing countries [41].

Corbett and Kirsch in 2001 [42] were the first authors to report on diffusion models of management systems (MSs), and recent models were reported by Cabecinhas et al. in 2018 [43] and Cabecinhas et al. in 2020 [44] regarding European countries and the diffusion of integrated management systems. Analyzing the global spread of ISO 14001, Corbett and Kirsch [42] stated that the behavior of the international environmental management standard certification is strongly correlated with the behavior of ISO 9001 certification. Revisiting ISO 14001 diffusion, Vastag [45] evidenced that the installed base of ISO 9001 certificates has a significant contribution to ISO 14001 certification density. The author considered the overlap between the two ISO standards and the convenient infrastructure, since ISO 9001 certification affords a relatively low-cost route to corporate image improvement and competitive advantage.

Based on the study authored by Cabecinhas et al. [44], Figure 2 summarizes in a timeline (from 2007 to 2020) the latest countries and standards studied in the domain of standardized management system diffusion.

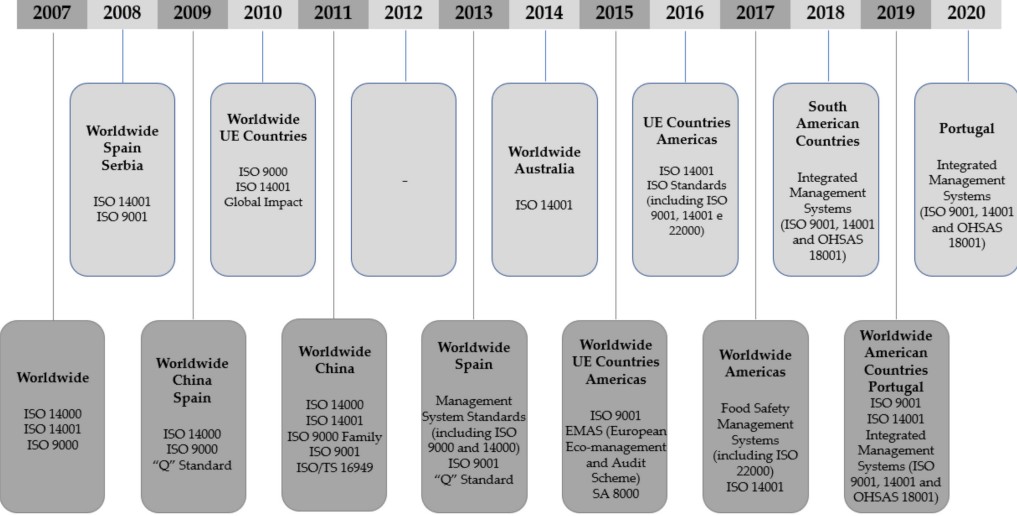

**Figure 2.** Timeline from 2007 to 2020 of the latest countries and standards studied in the domain of standardized management system diffusion, based and adapted from [44].

Analyzing the evolution of worldwide ISO 9001 diffusion according to activity sectors, Llach et al. [46] found that wholesale and retail sale are considered in expansion due to the mid-point on the grown curve suggested by the forecasting model, with a high growth rate. Regarding retailers involved in food supply chain operations, they have been seen

as a vital change agent towards sustainable food systems [47]. Therefore, the adoption of standards, including sustainable standards, is a necessity since greening the supply chain is a concern for businesses and a challenge for logistics management in the 21st century [48].

Several studies have described the phenomenon of standard diffusion by testing models with S-shaped behavior ([43,49–52]), which initially properly shaped data on bio-population growths and process of implementation of new technologies ([53,54]). Curves representing a process of diffusion must be an elongated S-shape, also called as a sigmoidal curve [55].

In terms of the growth of a bio-population, Buchanan et al. [56], Franceschini et al. [54], and Martino [57] described different phases to explain the behavior of the curve in standards diffusion.

Lag phase: the beginning of the diffusion of MS certification. The initial difficulties of the implementation must be overcome by organizations.

Exponential phase: a subsequent increase in the rate, producing a "steep" gradient to the curve [58]. After overcoming the initial difficulties of the implementation, the companies seek the certification, attracted by the benefits derived from the implementation of the standards [59].

Stationary phase: Also named maturity stage, the evolution of the number of certifications keeps constant and achieves the saturation effect [60].

Decline phase: Franceschini et al. [59] and Marimón et al. [51] reported on this fourth stage encompassing the decertification stage.

Related to the above, Marimón et al. [49] classified the countries into three types of standard expansion behaviors: expansionist, mature, and retrocessive. Analyzing the diffusion of ISO 9001 and ISO 14001 in European countries such as Spain and Serbia, these authors concluded that some countries in Central and Eastern Europe (e.g., Bulgaria, Hungary, Serbia, and Estonia) were the proper representatives of expansionist countries, whereas Spain was close to the maturity stage. According to Franceschini et al. [60], when the diffusion reaches the saturation point, certification becomes less attractive.

In addition to those aforementioned phases, Mastrogiacomo et al. [61] also suggested a new last diffusion phase named "post-decline," in which the number of certifications tends to stabilize at a lower level. However, the theorization of this last phase represents a novelty that updates the existing literature; thus, future studies must clarify the drivers and variables affecting it.

Considering well-known S-shaped curves in population dynamics, the law of Benjamin Gompertz [62] is a less-known example of population dynamic analysis, possibly due to its double exponential approach and its asymmetry [63]. Gompertz's paper is considered an important influence to form the basis for actuarial science and, nowadays, is very well known to practicing actuaries [64].

Following the development by Winsor [65] of Gompertz's equation, the Gompertz model is a basis of growth curves, both for biological and economic phenomena. Winsor wrote Equation (1), in which $k$ and $b$ are essentially positive quantities. The author showed that the $x$-value becomes positively infinite and that $y$ will approach the $k$-value.

$$y = k \times e^{-e^{a-bx}} \tag{1}$$

## 3. Materials and Methods

Data on ISO 22000 were collected from the ISO Survey of Certifications (2020) and ISO Survey of Certifications (2018). Each year ISO publishes on its website data concerning ISO standard certification all over the world. Information reported in these surveys is widely used in diffusion studies (see [43,51,66]). In this study, the number of ISO 22000 certificates issued in Europe from 2007 to 2018 was collected.

Table 2 presents the raw data collected and adopted to populate the forecasting model.

**Table 2.** Available data used to construct the forecasting model.

| Year | Counter | ISO 22000 Certificates Issued |
|------|---------|-------------------------------|
| 2007 | 1 | 2749 |
| 2008 | 2 | 4865 |
| 2009 | 3 | 6050 |
| 2010 | 4 | 7083 |
| 2011 | 5 | 7361 |
| 2012 | 6 | 8307 |
| 2013 | 7 | 9357 |
| 2014 | 8 | 10,181 |
| 2015 | 9 | 11,181 |
| 2016 | 10 | 11,083 |
| 2017 | 11 | 10,342 |
| 2018 | 12 | 9655 |

It must be highlighted that the available data on ISO 22000 certification are not enough to evaluate the diffusion process. Hence, for the next few years of ISO 22000 diffusion in Europe, data were extrapolated from the growth of ISO 9001 and ISO 14001, i.e., estimated from the ISO 9001 Survey (2020), ISO 14001 Survey (2020), and ISO Survey (2018). Table 3 presents the amount of data extrapolated and adopted to develop the forecasting model.

**Table 3.** Data extrapolated to construct the forecasting models.

| Year | Counter | ISO 22000 Certificates Estimated | |
|------|---------|-------------------------------|---------------------------|
| | | Based on ISO 9001 Growth | Based on ISO 14001 Growth |
| 2019 | 13 | 11,405 | 9472 |
| 2020 | 14 | 12,545 | 10,514 |
| 2021 | 15 | 13,047 | 10,934 |
| 2022 | 16 | 13,830 | 11,262 |
| 2023 | 17 | 15,213 | 11,375 |
| 2024 | 18 | 16,126 | 11,489 |
| 2025 | 19 | 14,029 | |
| 2026 | 20 | 14,310 | |
| 2027 | 21 | 14,024 | |
| 2028 | 22 | 13,883 | |
| 2029 | 23 | 13,467 | |
| 2030 | 24 | 13,871 | |

The selection of these ISO management standards was supported by the strong correlation between the aforementioned standards (as pointed out by [42]) and also based on their successful dissemination throughout Europe. This region accounts for 40% of ISO 9001 and 33% ISO 14001 certificates issued all over the world (ISO Survey, 2019).

Historically, these results evidence the huge relevance that Europe holds related to the worldwide diffusion of ISO 9001 certification. Therefore, Europe is the geographical zone where ISO 9001 certification began the earliest and the majority of these countries share common social and economic characteristics, including open markets and access to workforce [67].

The current study is supported by the methodology adopted by Cabecinhas et al. [44]. The growth curve adopted was the Gompertz model due to its wide use in scientific literature [68] and due to the fact that the Gompertz curve most accurately describes the dynamics of regions that have not reached the maximum saturation point [43].

Supported by several studies that adopted the Gompertz model ([55,69,70]), Cabecinhas et al. [43] considered the equation below as the differential equation describing the Gompertz model (2), where $k$ accounts for the time when the point of inflection is reached:

$$\frac{dy}{dt} = k \times y \times \ln\left(\frac{a}{y}\right) \qquad (2)$$

This study adopted the "SGompertz" function from the "growth/sigmoidal" category of the software OriginPro® 2020. Considering the solution of the Gompertz model in Equation (3), Cabecinhas et al. [44] considered $a$ the saturation value, which accounts for the maximum number of certificates that potentially will be issued.

$$y(t) = a \times e^{-e^{[-k \times (t - t_c)]}} \tag{3}$$

It is important to emphasize that the number of available data and the inclusion of the inflection point in the growth curve directly affect the results [71]. To calculate the point of inflection on the $y$-axis, Winsor [65] considered Equation (4), or when approximately 37% of the final growth had been reached. The asymmetry of the point of inflection is the characteristic of this model [72].

$$y = k/e \tag{4}$$

## 4. Results and Discussion

In 2015, 11,181 ISO 22000 certificates were issued in Europe (ISO 22000 Survey, 2020). The forecasting models were developed using the OriginPro® software encompassing the data in Tables 2 and 3. The parameters of each model, such as the equation and $a$-value, are reported in Tables 4 and 5. The "counter" value was considered an independent variable.

**Table 4.** Parameters and statistics for the Gompertz fitting of available European data.

| Curve Parameters | |
|---|---|
| $a$ | 11,774 |
| $xc$ | 1.90538 |
| $k$ | 0.29778 |
| $R^2$ | 0.97086 |

**Table 5.** Parameters and statistics for the Gompertz fitting of extrapolated European data.

| Curve Parameters | Based on ISO 9001 Growth | Based on ISO 14001 Growth |
|---|---|---|
| $a$ | 14,873 | 10,996 |
| $xc$ | 2.8585 | 1.6913 |
| $k$ | 0.16797 | 0.343 |
| $R^2$ | 0.92749 | 0.94322 |

Considering the statistics parameters, it is possible to point out that they presented a good fit. Figures 3–5 depicts a graphical representation of the data.

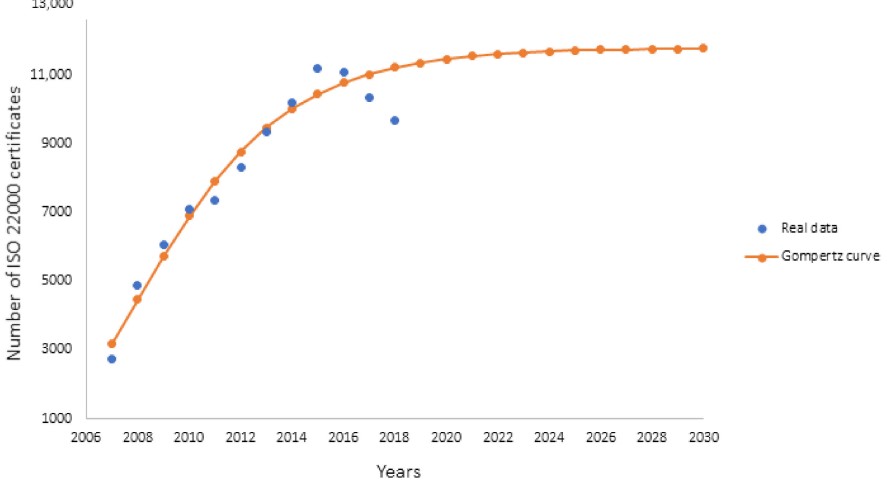

**Figure 3.** ISO 22000 diffusion based on available data on ISO 22000 certification.

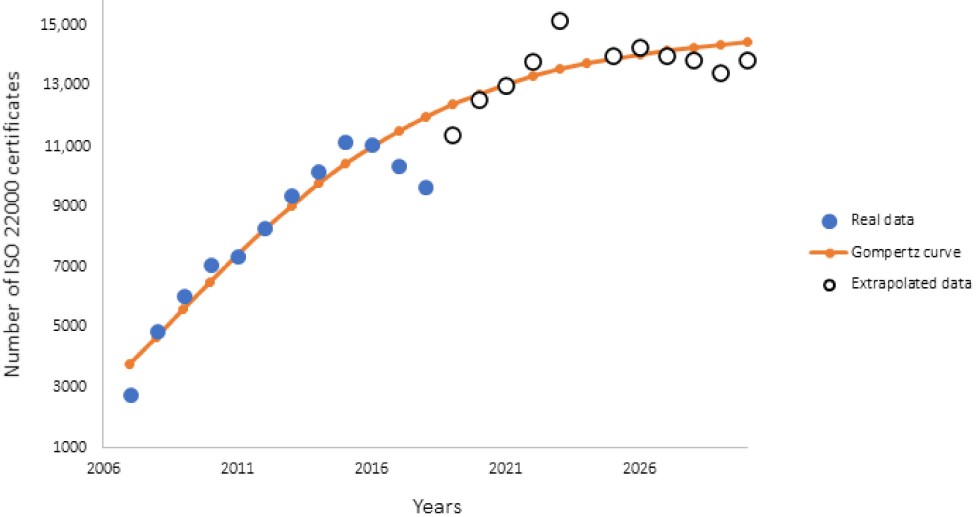

**Figure 4.** ISO 22000 diffusion based on ISO 9001 growth.

### 4.1. Results of the Forecasting Model without Extrapolated Data

Considering the forecasting model based on the available data on ISO 22000 certification (Figure 3), it is possible to state that Europe achieved the inflection point in 2008, reaching a total of 4329 certificates issued and a saturation level of 11,774 certifications, approximately.

According to the above, ISO 22000 certification could decline since the adoption of standards that reached the saturation stage is considered less attractive [60]. Regarding ISO 9001 and ISO 14001 diffusion, Marimón et al. [49] concluded that the intrinsic value of certification of those standards tends to decrease when the certificate is not a distinguishing factor.

However, Europe probably did not achieve the saturation level because the available data were not enough to evaluate only the diffusion process. The results of ISO 22000 certification behavior in European countries based on only available data include uncertainties, and to figure out the lack of data on ISO 22000 certification, it is necessary to update it with new data from the forthcoming years.

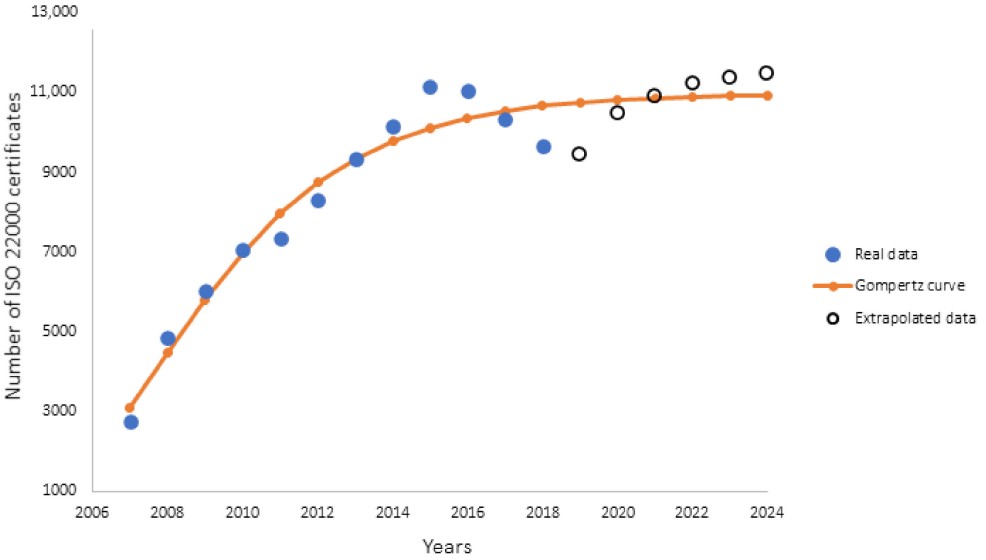

**Figure 5.** ISO 22000 diffusion based on ISO 14001 growth.

*4.2. Results of Forecasting Models with Extrapolated Data*

The results of both forecasting models indicate that they reach the same level of growth and suggest that Europe is positioned in the stationary phase. However, observing the $R^2$ value, it is possible to conclude the forecasting model based on the ISO 14001 growth more properly describes the current diffusion process. Furthermore, considering the *k*-value as the growth rate, it is possible to assume similar findings since the forecasting model based on ISO 14001 growth reached the point of inflection in approximately 34% of the final growth curve.

Analyzing the ISO 22000 dynamic of growth based on ISO 9001 (Figure 4), it is possible to state that Europe achieved the inflection point in 2009 with a total of 5468 certificates issued and reaching a saturation level of 14,873 certifications (approximately), whereas based on ISO 14001, the inflection point was attained in 2008 with a total of 4043 certificates issued and reaching a maximum of 10,996 certifications (Figure 5).

Some uncertainties were ascribed to the results by estimations and extrapolations introduced in the methodology applied since the amount of provided information assured a good performance of the forecasting model [71].

It was observed that the behavior of ISO 22000 certification based on the available real data was similar to the dynamics of diffusion of ISO 22000 based on ISO 9001 and ISO 14001 growth. Furthermore, this result corroborates the similarities that were found in the forecasting model of ISO 22000 based on ISO 14001 growth, since both proposed forecasting models reached the point of inflection in 2008 and represented 30–34% of the final growth curve.

The difference between the two dynamics of ISO 22000 diffusion suggests maximum and minimum intervals related to the number of certificates issued in Europe (Figure 6) since similarities among ISO 9001 and ISO 14001 diffusion behaviors were pointed out by some studies [73].

Considering the forecasting models encompassing extrapolated data, ISO 22000 certification in Europe will achieve a maximum of 14,873 and a minimum of 10,996 issued certificates if the dynamic of ISO 22000 diffusion behaves like ISO 9001 and ISO 14001 growth, respectively.

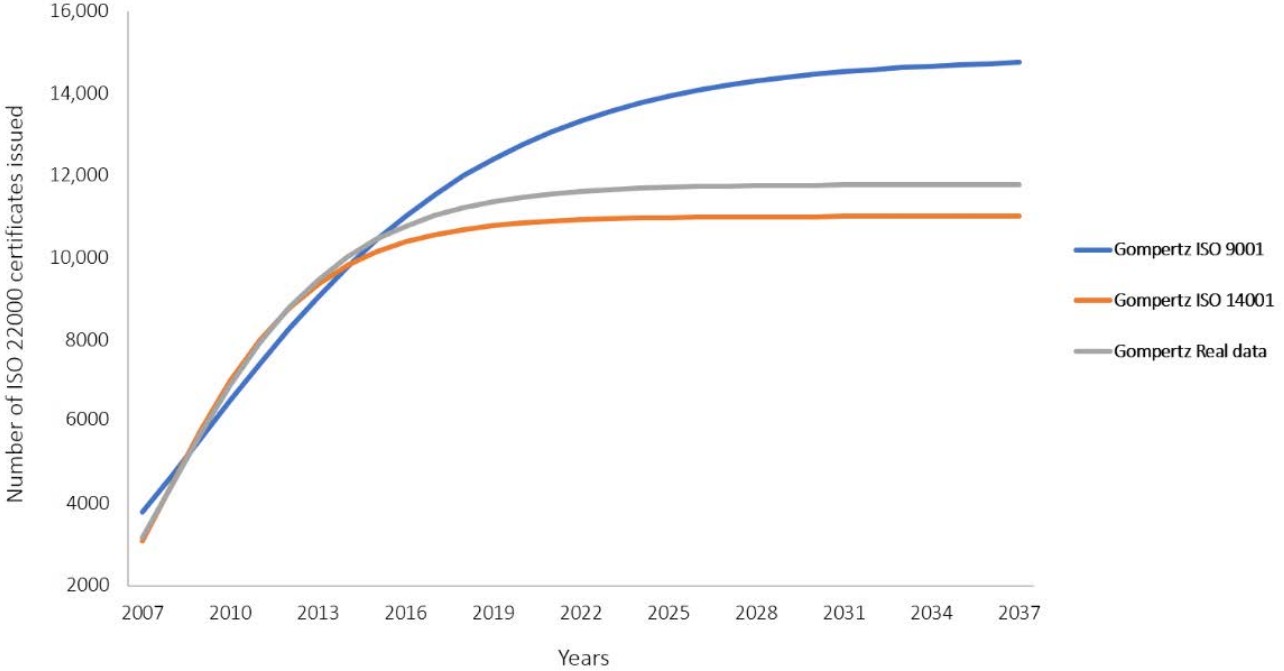

**Figure 6.** Suggested maximum and minimum intervals related to ISO 22000 certificates issued in Europe based on the forecasting model with extrapolated data from ISO 9001 and ISO 14001 growth.

Although the uncertainty predicted is due to the explained limitations, it is possible to assume that the process of diffusing the ISO 22000 standard in Europe is positioned near the minimum interval suggested. This corroborates the forecasting model without extrapolated data, in which ISO 22000 certification in Europe will reach a maximum of 11,774 certificates issued.

According to Figure 6, it is possible to stress that the saturation level of the forecasting models based on real data and on extrapolated ISO 14001 data will probably be achieved between the years 2019 and 2022 (supposing the data follow a similar pattern). Conversely, the forecasting model supported by the extrapolated ISO 9001 data suggests that the saturation level will be achieved between the years 2034 and 2037.

It must be highlighted that the uncertainty associated with those empirical results will be minimized by updating new data from forthcoming years. There is the possibility that these intervals have not yet been reached since the ISO 22000 standard is not popular in European countries [16] and the diffusion of this standard is taking place at a time marked not only by the economic crisis, but also by the coexistence with other food system certifications, such as BRC and IFS [29]. In addition, it is necessary to emphasize that the internationally recognized FSSC 22000 is a complete certification scheme for FSMS based on existing standards for certification, such as ISO 22000 (FSSC 22000 (2020)), thus, it has a major influence on which FSMS standards will be required in the next few years in Europe.

Considering that global agriculture and food trade are governed by various standards and regulations [37], food retailers play an important role in the emergency and diffusion of global food safety standards [74]. The poor popularity of the ISO 22000 standard is probably related to the adoption of standards that originate from retail actions, which are required by several large retailers, wholesalers, and foodservice companies. Considering the study carried out by Havinga [74], the majority of certified European organizations are distributed over three food safety standards. First, GlobalG.A.P. is the dominant food safety standard used in Europe. Second is BRC and third is FSSC 22000. This reinforces that geography provides market power to the standard holders [20] since all three aforementioned standards are European.

In addition to the above, Zimon and Domingues [16] suggested that the ISO 22000 standard is not widely implemented based on studies that claim that many companies do not see or seek its potential ([29,75,76]). Moreover, to define a standard adoption, the companies also need to realize benefits and costs related to the specific standardization process [77] and, even though many companies from European countries recognize the positive effects of ISO 22000 implementation on the supply chain [44], there is no framework to facilitate the understanding of factors that influence the adoption of the standard [78].

In a study related to reasons and constraints of implementing the ISO 22000 standard, the authors Escanciano and Santos-Vijande [29] dealt with the potential constraints that may difficult the implementation and certification of ISO 22000. The main constraints to the diffusion and application of this standard concern the high cost, lack of knowledge (unfamiliar), and the fact that ISO 22000 certification is not currently seen as a prerequisite for doing business. These factors emphasize that food businesses still do not understand this international standard and its potential.

Furthermore, all of the mentioned food safety standards are used by different and specific segments of the food chain, aiming at different food business operators. Thus, ISO 22000 standard implementation and its diffusion may be affected because of its applicability to all organizations in the food and feed industries, regardless of size or sector, as well as the non-recognition by GFSI.

## 5. Conclusions

This research paper presents the first attempt to empirically analyze the dynamic of the diffusion of the European ISO 22000 certification based on available real data and also based on the growth of renowned standards, namely, ISO 9001 and ISO 14001.

The forecasting based on ISO 9001 growth suggests that the inflection point was reached in 2009, and predicted a maximum of 14,873 certifications. On the other hand, the inflection point was reached in 2008 considering both the forecasting based on ISO 14001 growth and without extrapolated data, predicting a maximum of 10,996 and 11,771 certifications, respectively.

Based on the forecasting models developed, the intervals of the estimated number of ISO 22000 certificates issued in Europe was proposed. Concerning the forecasting model with extrapolated data established by using the Gompertz model, Europe is positioned in the stationary phase and will range from 14,873 as the maximum and 10,996 as the minimum of certificates issued if the dynamic of ISO 22000 diffusion behaves like ISO 9001 and ISO 14001 growth, respectively.

However, the results of the forecasting model without the extrapolated data suggest that Europe reached the saturation stage. The evolution of the number of certificates may keep constant and lead the diffusion process to the next stage, the decertification phase.

The approach adopted in this study presents some limitations. There is the possibility that these intervals have not yet been reached due to the lack of available real data regarding ISO 22000 adoption. The data concerning ISO 22000 certification is only available for the period from 2007 to 2018, and this fact prevents the assessment of the diffusion process. Data from ISO 9001 and ISO 14001 diffusion were extrapolated in order to overcome this shortcoming.

Overall, the forecasting models developed in this study hold important implications for future research. From the political point of view, the results of this study could support planning political activities and design regulations regarding globalization and international trade. Furthermore, it could help companies to coordinate their activities with the global requirements, which are increasingly orientated towards internationally certified products.

Lastly, food losses and food waste are considered a global issue, and to emphasize this, recent global events evidence that our current food production is unsustainable. Deeper research projects are fundamental to understand the processes of diffusion of food safety management system standards.

Although the statistic parameters of these models present a good fit, as reported before, there is a lack of information concerning ISO 22000 certification and, consequently, its diffusion. However, with further information available in the forthcoming years, better fitting with real results may be expected.

Lastly, in order to have a better idea of the real meaning of the estimates presented, future work may also include an analysis of important variables, such as the number of food companies that are eligible to be certified by ISO 22000 and the factors that are driving this behavior specifically in Europe.

**Author Contributions:** Conceptualization, P.D.; formal analysis, N.G.; writing—original draft, N.G.; writing—review and editing, P.D., M.C., D.Z. and P.S. All authors have read and agreed to the published version of the manuscript.

**Funding:** This research was funded by FCT—Fundação para a Ciência e Tecnologia within the R&D Units Project Scope: UIDB/00319/2020 and FCT Doctorate Grant Reference SFRH/BD/131932/2017.

**Informed Consent Statement:** Not applicable.

**Data Availability Statement:** Publicly available datasets were analyzed in this study. This data can be found here: https://www.iso.org/the-iso-survey.html.

**Conflicts of Interest:** The authors declare no conflict of interests.

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
