# Peer review of "ISO 22000 Certification: Diffusion in Europe"

_resources, doi:10.3390/resources10100100_

Round 1
Reviewer 1 Report
The purpose of this paper is to analyze the diffusion of implementing ‘Food Safety Management Systems’ (FSMS) in the European continent by using forecasting models. In my opinion, this paper can be accepted for publication after the following revisions:
- Suggest rewriting the “Abstract” to briefly describe the motives, purposes, research method, important results and implications, limitations and future research directions of this research.
- Suggest enhancing the description of the motive of this research in the “Introduction” section.
- Suggest enhancing the descriptions of implications of the research results.
- Suggest enhancing the description of the contribution of this paper in the field.
- Suggest citing the related papers published in this journal “Resources” in the recent two years to link this paper to the material of this journal (only three among 76 cited references).
Author Response
Thank you for your contribution. The responses were uploaded below.

Reviewer 2 Report
The reasearch article on “ISO 22000 Certification: Diffusion in Europe” globally describes food safety management arrangements among food chain organizations in accordance with ISO 22000. The paper is very introductory to be considered a research article and the results seem not to be connected among them; also, the link within ISO 9001 and 14001 should be better clarify. Furthermore, the research question and the objective of the study is confusionary. Also the nexus of the results obtained with the application of the “Gompertz function” do not underline the linkage with food losses and waste. The paper should be implemented with more concrete and effective case-study, also considering the aspect of food losses and waste.
Below, here are also some errors to check in the text:
Lines 17-18: the sentence “Data concerning the evolution of ISO 22000 certificates issued was collected for the period between 2007 and 2019” is unconnected with the figure 2, where the period is extended to 2020.
Lines 29-30: sentences are not clear. Try to clarify the concept
Line 49: the sentence “until these products to come home of the consumers” is not clear. English language check is required.
Lines 137-138: the sentences are not clear. You should better describe it!
Line 203: Was the ISO 22000 food safety management system only applied in “spaghetti industry”?
Lines 205-219: the concept expressed in these lines seems to be unlinked. You should try to focus on the objective of the research article and well argument it!
Line 350: there is an error: “bellow”
The manuscript should also be edited in the format of the journal.
Author Response

(The authors gave the same response as above.)

Reviewer 3 Report
Manuscript ID: resources-1346406
Title: ISO 22000 Certification: Diffusion in Europe
Article Type: Research article
The authors propose an analysis of the diffusion of implemented Food Safety Management Systems (FSMS), considering the ISO 22000 standard in Europe.
This is an interesting and original analysis, still, the construction of the text and phrasing are sometimes confusing.
The introduction does not address ISO 22000 in relation to FSSC 22000, when this is a real issue in the food industry, as most of the certified companies are transitioning and requiring an upgrade to FSSC 22000 - it must be properly addressed, as this has a major influence on which FSMS standards will be required in the next few years in Europe and why.
The literature review is complete and well written, and the material and methods, as well as the results and discussion sections are well written and need minor revision.
The conclusion needs a thorough revision.
General comment:
- i) Revision by a native English speaker is required before publication.
- ii) Improve the abstract and introduction sections, allowing it to adequately present the manuscript.
iii) Downtune the conclusion.
Specific comments:
Page 1, line 28: “Related to these consumers, the enterprise’s concerns associated with new requirements from stakeholders have increased over the last years.” What is this sentence supposed to mean?
Page 1, line 48: “since food products are produced using different technics of cultivation and manipulation and transported all over the world until these products to come home of the consumers”. This is another sentence that seems to miss some words, ending up confusing to the reader. please rephrase.
Page 1, line 51: “To improve food safety, a Food Safety Management System (FSMS) standard based on good manufacturing practices and HACCP (Hazard Analysis and Critical Control Point) was published in 2005 by International Organization for Standardization”. How does the publication of a standard improve food safety? I get the idea, but the authors seem to lack basic knowledge on the ISO jargon and purpose.
Page 1, line 53 “ISO 22000 is an international standard that requires a
food safety management system’s implementation to assure food security.” . What do the authors mean? Food safety and food security are different concepts – the authors need to explain how does the FSMS implementation assures food security, as I believe this must be a misconception.
Page 2, circa line 65 “When reviewing the literature, there is a lack of research and studies covering this subject.” this is a first sentence of an entire paragraph. the former paragraph addresses the necessity of standards supporting the care of food quality. I cannot understand to which subject are the authors referring to. Please clarify.
Page 2, line 71 “The main goal of this work is to answer the following research questions: is the Gompertz model suitable for studying the diffusion of ISO 22000 in Europe? Please include this aim in the abstract.
Page 5, line 248 “his infrastructural convenience” Please review.
Page 7, line 317 In the materials and methods section, the authors could have estimated based on available data the number of food companies that are eleictible to be certificated by ISO 22000, in order to have a better idea on the real meaning of the estimates presented in Results and discussion sections.
Page 9, line 414 to 417: Can this decline be due to the FSSC 22000 preference by food companies?
page 12, line 530-532: The authors need to explain that these food safety standards are used by different segments of the food chain, aiming different food business operators. This is not a justification for what has been presented before. Also, the authors should elaborate on the reasons for choosing a standard, and one of the main reasons is related to clients demand and the geographic location and dispersion of this client (retailers) has a lot to do with it. This has to be properly assessed and discussed.
page 12, conclusion: this section needs to be downtuned, addressing the real conclusions of the work.

Author Response

(The authors gave the same response as above.)

Round 2
Reviewer 2 Report
According to the reviewer comments the modification have been done. However, the manuscript seems to be still scarce, thus treating the argument as a review and in a summarized way. I think that you should improve it with more practical cases, in order to consider it as a research article.
As reference to what has been written you should implement the paper also considering these references:
- Fathurrahman, R. N., Rukayadi, Y., Ungku Fatimah, U. Z. A., Jinap, S., Abdul-Mutalib, N. A., & Sanny, M. (2021). The performance of food safety management system in relation to the microbiological safety of salmon nigiri sushi: A multiple case study in a japanese chain restaurant. Food Control, 127 doi:10.1016/j.foodcont.2021.108111
- Chen, H., Liou, B. -., Hsu, K. -., Chen, C. -., & Chuang, P. -. (2021). Implementation of food safety management systems that meets ISO 22000:2018 and HACCP: A case study of capsule biotechnology products of chaga mushroom. Journal of Food Science, 86(1), 40-54. doi:10.1111/1750-3841.15553
- Zimon, D., Madzik, P., & Domingues, P. (2020). Development of key processes along the supply chain by implementing the ISO 22000 standard. Sustainability (Switzerland), 12(15) doi:10.3390/su12156176
Reviewer 3 Report
Dear Authors,
I have analyzed the revised version of the manuscript and agree with this version, but still feel that it needs to be revised by a native English speaker.
